# Effect of Dry–Wet Cycle Periods on Properties of Concrete under Sulfate Attack

Jin-Jun Guo [1], Peng-Qiang Liu [1], Cun-Liang Wu [2] and Kun Wang [1,*]

[1] School of Water Conservancy Science and Engineering, Zhengzhou University, Zhengzhou 450001, China; guojinjun@zzu.edu.cn (J.-J.G.); pqliu@gs.zzu.edu.cn (P.-Q.L.)
[2] China Construction Seventh Engineering Division Co., Ltd., Zhengzhou 450001, China; WCLiang@126.com
[*] Correspondence: wangkunli@gs.zzu.edu.cn

**Abstract:** Dry–wet cycle conditions have significant effects on the corrosion of concrete under sulfate attack. However, previous studies have only applied them as a method for accelerating sulfate attack and not systematically studied them as an object. In order to explore the impact of sulfate attack with different dry–wet cycle periods on concrete, in this study, four dry–wet cycle periods (3, 7, 14, and 21 days) were selected. The flexure strength, relative dynamic modulus, and mass were tested, and the microstructures of the eroded specimens were also analyzed. The intensity and depth of sulfate erosion were influenced by the wet–dry cycle period. The results show that the deterioration of concrete first increased and then decreased with an extension of the dry–wet cycle period. Microstructural analysis indicated that, with an increase in the dry–wet cycle period, the corrosion depth of sulfate attack increased. Moreover, the erosion products such as ettringite and gypsum were greatly increased, in agreement with the macroscopic variations. However, excessively prolonging the dry–wet periods does not significantly further the deterioration of concrete's performance. Therefore, considering the strength and depth of corrosion caused by sulfate attack, it would be appropriate to employ dry–wet cycle periods of 7–14 days under natural dry conditions in studies on concrete.

**Keywords:** sulfate attack; dry–wet cycle period; concrete; erosion intensity

## 1. Introduction

Sulfate attack is one of the most important factors that affect the durability of concrete, and it is greatly accelerated by dry–wet cycles [1–3]. Sulfate erosion mainly includes two aspects: physical and chemical erosion [4]. The first occurs through the crystallization of sulfate, producing crystallization pressure and cracks in concrete structures [5], while chemical erosion is mainly caused by expansive products, such as ettringite (AFt) and gypsum, produced by the reaction between sulfate and cement hydration products. These products make concrete expand, crack, flake, disintegrate, and increase in permeability, thus providing access for other erosive media (e.g., $Cl^-$ and $CO_2$) to enter the interior of the concrete, resulting in the acceleration of its deterioration. Simultaneously, the erosion process consumes much OH-, leading to the decomposition of the unstable hydrated calcium silicate gel (C-S-H), thereby weakening the bonding effect and reducing the strength of the concrete [6,7].

In environments presenting dry–wet cycles, such as those exposed to splashes and tides, the sulfate attack of concrete is very complicated [8]. During the wet process, $SO_4^{2-}$ gradually penetrates into the concrete through microcracks and pores and accumulates. During the dry process, the water rapidly evaporates, and the concentration of $SO_4^{2-}$ in the concrete pores gradually increases, while $Na_2SO_4$ crystallizes [9]. The $SO_4^{2-}$ reacts with the cement hydration products, which increases the rate of expansion and cracking, eventually accelerating the deterioration of the concrete [2]. Chen et al. [10] and M. Sah-

maran et al. [11] analyzed the effects of sulfate attack and dry–wet cycles on cement-based materials, showing that dry–wet cycles significantly influenced the sulfate attack process.

In recent years, many researchers have studied new materials or methods to enhance the ability of concrete to resist sulfate attack under dry–wet cycles. Some have studied the effects of different active admixtures (fly ash, slag, and silica fume) on concrete under sulfate attack in dry–wet cycles, finding the active admixtures to have positive effects on crack repair and the performance of concrete [3,12,13]. Yang et al. found that a biofilm coating could effectively improve the resistance of concrete to sulfate attack [14]. Previous studies have shown the coupling of external load and sulfate attack under dry–wet cycles to negatively impact concrete. You et al. focused on the sulfate attack of concrete under a combination of bending, fatigue loads, and dry–wet cycles; the results showed that the loads and dry–wet cycles expedite the transport of sulfate ions in the concrete, accelerating the corrosion with sulfate [15]. The degradation of cement-based materials under the coupled effects of multiple corrosive ions and dry–wet cycles is another research direction. He et al. studied the effects on the transition zone of the concrete interface under dry–wet cycles with forms of sulfate attack, and the results suggest that the interface roughness increased with exposure time [2]. However, the dry–wet cycling regimes used in different studies on the erosion of concrete under the influence of sulfate attack and dry–wet cycles are unclear. This leads to differences between various studies, which is not conducive to integration and systematization for this area of research.

Over the past few years, some scholars have begun preliminary research into and exploration of the dry–wet cycle system. For example, Pang et al. divided the dry–wet cycle into two processes: drying and wetting. They studied the transfer of moisture in concrete and found that the drying time determines the depth of concrete deterioration [16] and the dry–wet cycle period significantly affects the depth of the wetting and drying process inside the concrete. Guo et al. [8] and Sutrisno et al. [17] tested the effects of different dry–wet duration ratios on concrete performance under sulfate or chloride attack, by taking the mechanical properties of concrete as evaluation indices. The results indicate that the degradation of concrete induced by attack increases and then decreases as the dry–wet ratio increases. Another research focus is simulating the actual environment to determine dry–wet cycle parameters. D.V. Reddy et al. designed dry–wet cycles of 12, 24, and 48 h to imitate coastal environments, such as in Iran [18]. Li et al. simulated the sulfate attack environment in Western China with a dry–wet cycle of 24 h and dry–wet ratio of 1:1 [19]. In order to simulate the marine tidal environment, Cheng et al. used a 24-h representation of a dry–wet cycle, soaking samples for 18 h and then drying them in air for 0.5 h, followed by drying the samples at 60 °C for 5 h, before finally cooling them at room temperature for 0.5 h [20]. Liu et al. used a 72-h dry–wet cycle and dry–wet ratios of 6:1, 9:1, and 14:1 to simulate alternating drying and wetting by the irregular, shallow, semi-diurnal tide at Lianyungang Port [21]. Comprehensive analysis showed that the main factors that need to be considered when testing dry–wet cycles are the drying and wetting duration, solution concentration, and temperature. In previous studies, most of the dry–wet cycle parameters were regarded as degradation conditions rather than objects of research. However, parameters such as the dry–wet cycle period have great influence on the deterioration of concrete under sulfate attack. To date, studies on the influence of the dry–wet cycle period on the long-term performance of concrete are insufficient for establishing a unified dry–wet cycle system standard, limiting the development and focus of research into concrete under sulfate attack.

It is therefore crucial to take the dry–wet cycle period as the main factor in establishing a reasonable dry–wet cycle method. This should facilitate systematic experimental studies on concrete's degradation in performance under sulfate attack in different dry–wet cycle systems and on the mechanism underlying it. Using different dry–wet cycle periods, this study evaluated the characteristics of concrete damage through sulfate attack based on several aspects such as the flexural strength, relative dynamic elastic modulus, and mass change. Considering that sulfate attack products might decompose when the temperature

is high, this study adopted the natural drying approach to avoid negative impacts from temperature on the dry–wet cycles [22,23]. The microstructural characteristics of concrete before and after sulfate attack and the crystal formation and morphology of the erosive products were analyzed by scanning electron microscopy (SEM) to reveal the mechanisms of damage under different dry–wet cycle periods and sulfate attack.

## 2. Experimental Program

### 2.1. Materials

Ordinary Portland PO42.5 Cement and fly ash (FA) were adopted in this study. Their main chemical components and physical properties are provided in Table 1. The XRD spectrum of the fly ash is shown in Figure 1, and Figure 2 shows the particle size distribution of the cement fly ash. The aggregate included natural river sand with a fineness modulus of 2.87 and basalt gravel with a continuous gradation of 5–20 mm in particle size. Replacing cement with 20% fly ash had been reported to significantly improve the resistance of concrete to sulfate attack [24–26]; thus, concrete with a strength grade of C30 and 20% fly ash were selected in this study. The mixture ratios are shown in Table 2. In addition, industrial-grade sodium sulfate with a purity of 99% was bought from a market. A test erosion solution was prepared with 5% sodium sulfate (by weight).

**Table 1.** Chemical composition and physical properties of cement and fly ash (%).

| Composition | Cement | Fly Ash |
|---|---|---|
| **Chemical (%)** | | |
| MgO | 3.34 | 1.5 |
| $SiO_2$ | 31.43 | 58 |
| CaO | 41.28 | 2.8 |
| $Al_2O_3$ | 12.43 | 30 |
| $Fe_2O_3$ | 3.34 | 4.3 |
| $K_2O$ | 0.80 | 1.36 |
| $SO_3$ | 3.22 | 1.22 |
| $Na_2O$ | 0.43 | 0 |
| LOI | 1.09 | 0.82 |
| $C_2S$ | 18.79 | - |
| $C_3S$ | 43.38 | - |
| $C_3A$ | 6.00 | - |
| $C_4AF$ | 8.55 | - |
| **Physical properties** | | |
| Spec. surf. area ($m^2$/g) | 332 | 287 |
| Density (kg/$m^3$) | 3.06 | 2.34 |
| Compressive strength at 3/28 days (MPa) | 26.6/54.5 | - |
| Flexural strength at 3/28 days (MPa) | 5.42/8.74 | - |
| Time of initial/final setting (min) | 90/300 | - |

**Table 2.** Designed mixture for concrete by proportion.

| W/B | Sand Ratio (%) | The Amounts of Different Materials (kg/$m^3$) | | | | |
|---|---|---|---|---|---|---|
| | | Water | Cement | Fly Ash | Sand | Aggregate |
| 0.54 | 36 | 195 | 289 | 72 | 664 | 1180 |

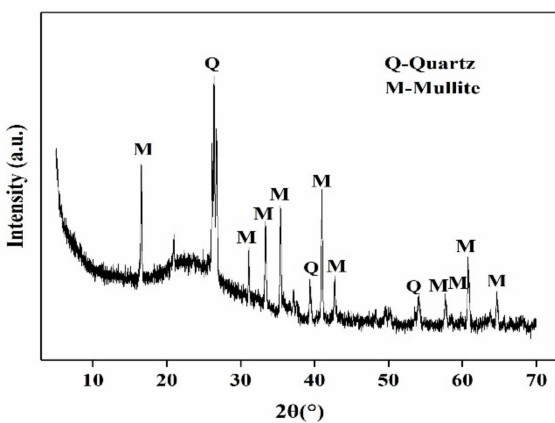

**Figure 1.** XRD spectrum for fly ash.

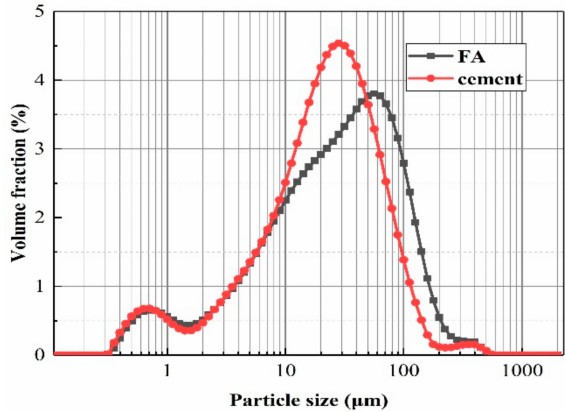

**Figure 2.** Particle size distribution of fly ash and cement.

### 2.2. Specimen Production and Maintenance Conditions

According to GB/T50081–2019 [27], a compulsory mixer was used to prepare concrete specimens with dimensions of $100 \times 100 \times 100$ and $400 \times 100 \times 100$ mm. For this purpose, the coarse aggregate, cementitious material, and fine aggregate were firstly added to the mixer separately and mixed for about 2 min; then, water was added and mixed in evenly; finally, the mixture was put into the mold and vibrated on a vibrating table until the concrete surface became a slurry. The specimens were molded for 24 h and then cured under standard conditions (temperature = $20 \pm 2$ °C; RH > 95%) for 28 days and labeled as T3, T7, T14, and T21.

### 2.3. Drying–Wetting Cycle Design

The dry–wet cycles were designed with reference to GB/T 50082-2009 [28] and ASTM C1012-18a [29]. Anhydrous sodium sulfate was used to prepare a 5% sodium sulfate solution as the test erosion solution. As shown in Table 3, this study employed four different dry–wet cycle periods (3, 7, 14, and 21 d) and a constant dry–wet duration ratio of 3:1. During the dry–wet cycle tests, the specimens were submerged in a polyethylene test box containing the sodium sulfate solution at room temperature as shown in Figure 3. During the drying process, the specimens were placed vertically upwards outdoors to dry naturally, maintaining good ventilation. The $Na_2SO_4$ solution was refreshed every 30 days to keep the solution concentration constant.

**Table 3.** Dry–wet cycle design.

| Code | Dry–Wet Cycle Period (Days) | Dry–Wet Ratio |
|:---:|:---:|:---:|
| T3 | 3 | |
| T7 | 7 | |
| T14 | 14 | 3:1 |
| T21 | 21 | |

Note: T7 means that the drying–wetting cycle period was 7 days, in which natural drying lasted 126 h and wetting lasted 42 h.

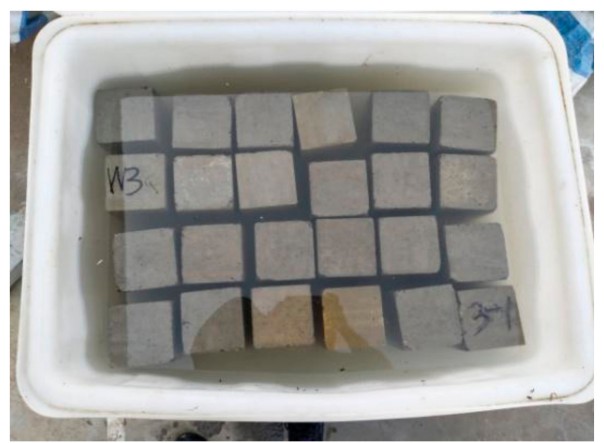

(**a**) Wetting process

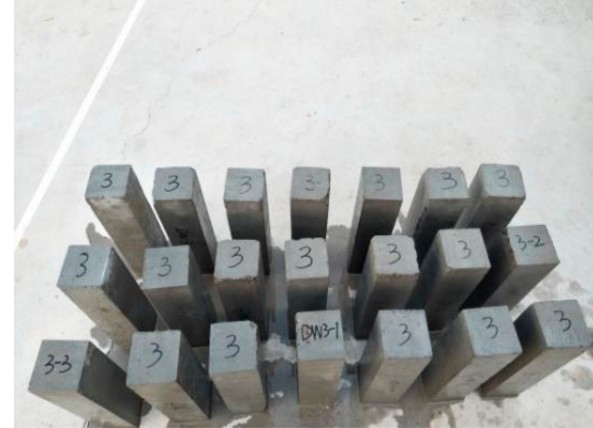

(**b**) Drying process

**Figure 3.** Wetting and drying process. (**a**) Wetting process; (**b**) drying process.

*2.4. Testing Procedure*

After a certain number of cycles, some properties of the specimens were tested, such as the flexural strength, dynamic elastic modulus, and mass change. The average values for the three specimens in each group were taken as the final values. The microstructure was then analyzed by scanning electron microscopy (SEM). The tests were carried out at regular intervals (approximately every 3 weeks) until 231 days.

2.4.1. Flexural Strength

According to GB/T 50081-2019 [27], the flexural strength was tested using a universal testing machine. The placement of the specimen is shown in Figure 4. The pressure-bearing surface of the specimen was the side, and the distance between the supports was 300 mm. The specimen was loaded at a rate of 0.06 MPa/s until broken, and then the failure pressure and fracture position were recorded. The flexural strength was calculated using Equation (1).

$$f_t = 0.85 \frac{Fl}{bh^2} \tag{1}$$

where $f_t$ = the flexural strength of the concrete (MPa), $F$ = the failure load of the specimen (N), $l$ = the span between pillars (mm), $b$ = the specimen width (mm), and $h$ = the height of the specimen (mm).

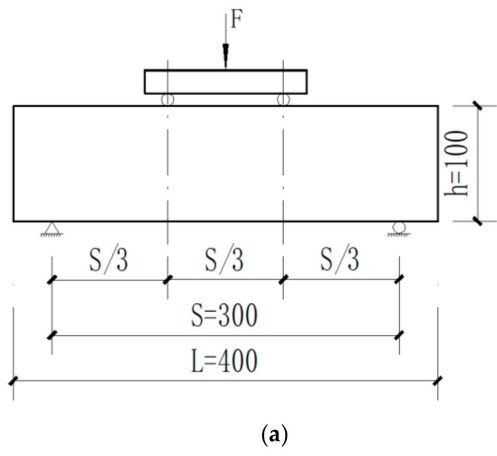

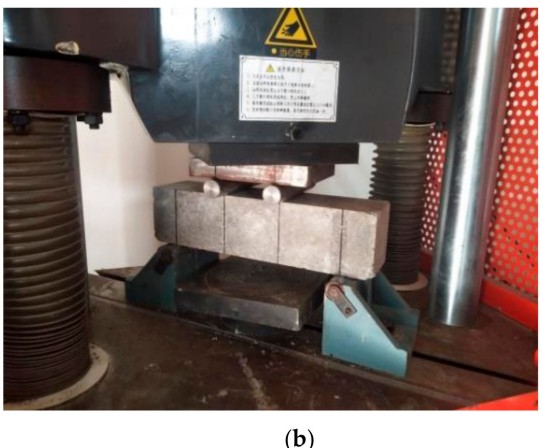

(**a**)                                                                   (**b**)

**Figure 4.** Flexural strength test for concrete. (**a**) Test schematic; (**b**) Test process.

The flexural strength of the sulfate-eroded concrete specimen over time was evaluated with the relative flexural strength coefficient $R_f$, as follows:

$$R_f = \frac{f_t^n}{f_t^0} \tag{2}$$

where $f_t^0$ and $f_t^n$ are the flexural strength after 0 and t dry–wet cycles, respectively.

### 2.4.2. Relative Dynamic Elastic Modulus

According to the Chinese standard GB/T 50082-2009 [26], the NELD-DTV instrument (Figure 2) was used to determine the dynamic elastic modulus, measuring in the middle of one side of the forming surface that had no visible holes or cracks. There were three specimens in each group, and the average dynamic elastic modulus was taken in each. The relative dynamic elastic modulus (RDEM ($E_{rd}$)) was used to analyze the erosion effect:

$$E_{rd} = \left( \frac{E_t}{E_0} - 1 \right) \times 100\% \tag{3}$$

where $E_0$ and $E_t$ are the dynamic elastic modulus after 0 and t dry–wet cycles, respectively.

### 2.4.3. Mass Change

The specimen's mass was measured using an electronic scale with an accuracy of 0.1 g. The average value for three specimens was recorded, and the mass-loss rate (Mc) for each group was calculated according to Equation (4):

$$M_C = \left( \frac{m_n}{m_0} - 1 \right) \times 100\% \tag{4}$$

where $m_0$ and $m_n$ are the mass after 0 and n dry–wet cycles, respectively.

### 2.4.4. Microstructural Analysis

The KYKY-EM6200 environmental scanning electron microscope (SEM) was used to characterize the microstructure and morphology of the concrete samples before and after sulfate attack with different dry–wet cycle periods. After specific periods of exposure, samples were taken from the failure specimens in the flexure strength tests at a certain depth and cut to appropriate sizes. Next, the samples were dried at 60 °C for 48 h and then sprayed with gold. Finally, the samples were put into the chamber and held under vacuum for testing.

## 3. Results and Discussion

### 3.1. Visual Inspection

During the wetting process, sulfate migrated into the specimen, and in the drying process, expanded sulfate crystals emerged in the pores of the specimen, resulting in the physical weathering of the exposed surface particles, which decomposed or fell off. Therefore, as the number of cycles increased, the surface of the specimen gradually chalked off, as shown in Figures 5 and 6. This is the prevalent salt-weathering effect observable in nature [30].

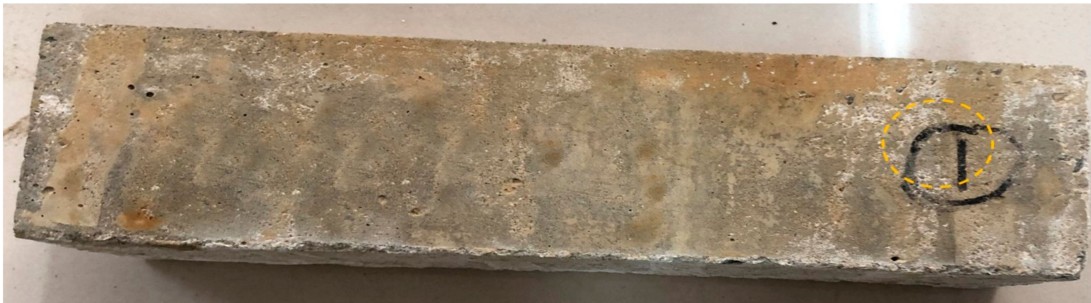

(**a**) Before sulfate erosion

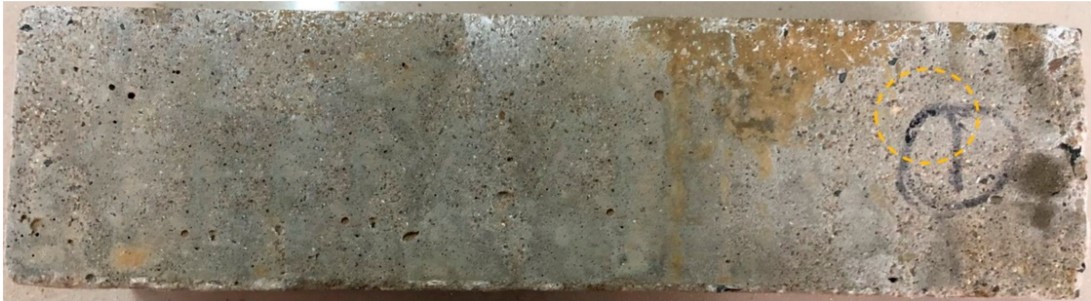

(**b**) After sulfate erosion

**Figure 5.** Visual inspection of concrete with dry–wet cycles. (**a**) Before sulfate erosion; (**b**) after sulfate erosion.

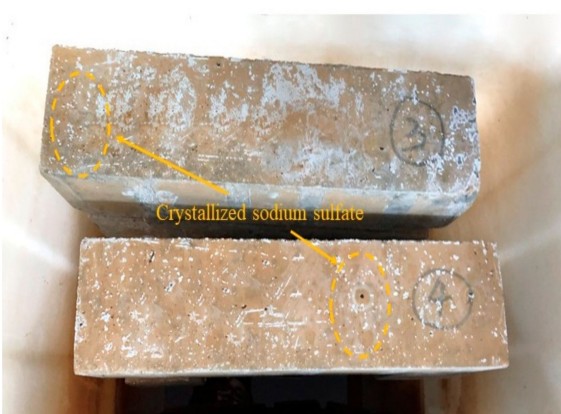

**Figure 6.** Crystallized sodium sulfate.

### 3.2. Flexural Strength of Concrete

The evolution of the relative flexural strength of the T3, T7, T14, and T21 specimens with the dry–wet cycles is shown Figure 7 and Table 4. It is apparent that throughout the attack process, the relative flexural strength did not change with different dry–wet cycle periods, but the damage and deterioration caused by a single cycle notably differed. The relative flexural strength of T3 peaked after 28 cycles, while that of T21 peaked after only

about four cycles. Therefore, the longer the dry–wet cycles, the great the relative flexural strength of the concrete, and the more obvious the effects of sulfate attack. Moreover, the variation in T14 and T21 was similar throughout the attack process, which shows that the excessive extension of the dry–wet cycle period had little effect on the flexural strength.

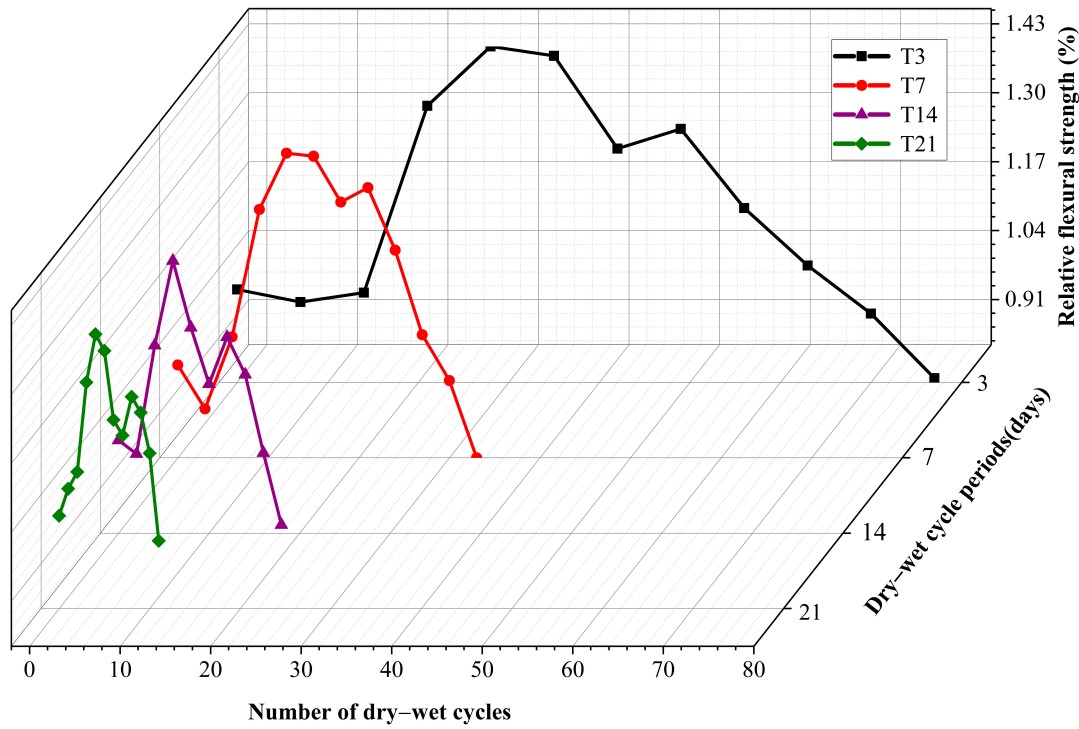

**Figure 7.** Variation of relative flexural strength among different dry-wet cycle periods.

After peaking, the relative flexural strengths in the four groups all rapidly decreased as the number of dry–wet cycles increased. The strength subsequently started to rebound at about 7, 10, 18, and 42 cycles, respectively, more so in T3 than the other groups. The rebound was caused by certain swelling products generated by the sulfate attack such as ettringite filling the concrete pores and cracks that resulted from the early attack [8,31]. At the same time, the hydrated calcium silicate gel (C-S-H) continued to dissolve, leading to the fluctuation in the flexural strength.

The single-attack intensity increased and the relative flexural coefficient decreased more quickly as the dry–wet cycle periods were increased in the terminal stage. The relative flexural strength of the concrete for T21 decreased the fastest: by 5.6% after each dry–wet cycle on average—higher than the 4.2% for T14, 2.7% for T7, and 1.05% for T3. T7 showed the largest general decrease; its relative flexural strength decreased by about 18% and 57% from the initial and peak values, respectively. The decreases for T14 and T21 were similar. This shows that the damage caused by sulfate attack with different dry–wet periods was not only related to the damage caused by a single cycle, but also related to the number of cycles. Therefore, both the degree of damage induced by a single dry–wet cycle and the number of cycles are important parameters determining the most unfavorable conditions for concrete.



**Table 4.** The flexural strength in the experiment (MPa).

| Cycles | T3 | | | Average Value | Relative Value |
|---|---|---|---|---|---|
| | **1** | **2** | **3** | | |
| 0 | 5.18 | 5.16 | 4.91 | 5.08 | 1.00 |
| 7 | 5.09 | 4.80 | 4.99 | 4.96 | 0.98 |
| 14 | 4.95 | 4.97 | 5.24 | 5.05 | 0.99 |
| 21 | 4.92 | 6.88 | 6.81 | 6.84 | 1.35 |
| 28 | 8.45 | 7.61 | 7.20 | 7.41 | 1.46 |
| 35 | 10.03 | 7.15 | 7.50 | 7.32 | 1.44 |
| 42 | 6.25 | 6.36 | 6.69 | 6.43 | 1.27 |
| 49 | 6.67 | 6.61 | 6.58 | 6.62 | 1.30 |
| 56 | 8.66 | 5.76 | 5.95 | 5.86 | 1.15 |
| 63 | 5.42 | 5.48 | 5.04 | 5.31 | 1.05 |
| 70 | 5.01 | 4.66 | 4.87 | 4.85 | 0.95 |
| 77 | 4.13 | 4.30 | 4.26 | 4.23 | 0.83 |
| | T7 | | | | |
| | 1 | 2 | 3 | | |
| 0 | 5.18 | 5.16 | 4.91 | 5.08 | 1.00 |
| 3 | 4.36 | 4.99 | 4.64 | 4.66 | 0.92 |
| 6 | 6.49 | 5.34 | 5.36 | 5.35 | 1.05 |
| 9 | 4.73 | 6.72 | 6.43 | 6.57 | 1.29 |
| 12 | 6.87 | 7.09 | 7.36 | 7.11 | 1.40 |
| 15 | 6.46 | 7.49 | 7.28 | 7.08 | 1.39 |
| 18 | 6.53 | 6.73 | 6.66 | 6.64 | 1.31 |
| 21 | 6.64 | 6.68 | 7.02 | 6.78 | 1.33 |
| 24 | 6.25 | 6.08 | 6.22 | 6.18 | 1.22 |
| 27 | 5.37 | 5.33 | 5.39 | 5.37 | 1.06 |
| 30 | 4.85 | 4.99 | 4.96 | 4.93 | 0.97 |
| 33 | 4.38 | 4.01 | 4.19 | 4.19 | 0.82 |
| | T14 | | | | |
| | 1 | 2 | 3 | | |
| 0 | 5.18 | 5.16 | 4.91 | 5.08 | 1.00 |
| 2 | 5.02 | 4.99 | 4.84 | 4.95 | 0.97 |
| 4 | 5.87 | 5.76 | 6.33 | 5.99 | 1.18 |
| 6 | 7.24 | 7.00 | 6.15 | 6.80 | 1.34 |
| 8 | 7.49 | 6.07 | 6.25 | 6.16 | 1.21 |
| 10 | 5.24 | 6.11 | 5.50 | 5.62 | 1.11 |
| 12 | 6.06 | 6.07 | 6.09 | 6.07 | 1.19 |
| 14 | 6.07 | 6.02 | 5.04 | 5.71 | 1.12 |
| 16 | 5.13 | 4.70 | 5.05 | 4.88 | 0.98 |
| 18 | 4.03 | 4.32 | 4.47 | 4.27 | 0.84 |
| | T21 | | | | |
| | 1 | 2 | 3 | | |
| 0 | 5.18 | 5.16 | 4.91 | 5.08 | 1.00 |
| 1 | 5.24 | 4.99 | 5.79 | 5.34 | 1.05 |
| 2 | 8.18 | 5.56 | 5.44 | 5.50 | 1.08 |
| 3 | 6.22 | 6.44 | 6.40 | 6.36 | 1.25 |
| 4 | 5.02 | 6.40 | 7.25 | 6.82 | 1.34 |
| 5 | 6.52 | 6.25 | 7.20 | 6.66 | 1.31 |
| 6 | 6.25 | 5.88 | 5.88 | 6.00 | 1.18 |
| 7 | 5.66 | 5.96 | 5.92 | 5.85 | 1.15 |
| 8 | 9.75 | 6.12 | 6.32 | 6.22 | 1.22 |
| 9 | 6.00 | 6.05 | 6.16 | 6.07 | 1.19 |
| 10 | 5.35 | 6.07 | 5.62 | 5.68 | 1.12 |
| 11 | 4.82 | 4.95 | 4.76 | 4.84 | 0.95 |

### 3.3. Relative Dynamic Elastic Modulus

The RDEM of the concrete varied with the number of dry–wet cycles (Figure 8 and Table 5). In general, the relative dynamic elastic moduli were similar between the groups, firstly increasing and then decreasing. However, the numbers of dry–wet cycles required to reach a given stage were different. For example, the longer the dry–wet cycle period, the fewer the cycles required to reach the peak value. In the initial stages, the RDEM in each group decreased slightly. At this point, the dry–wet cycles played a leading role, and the erosion effect of sulfate was weak.

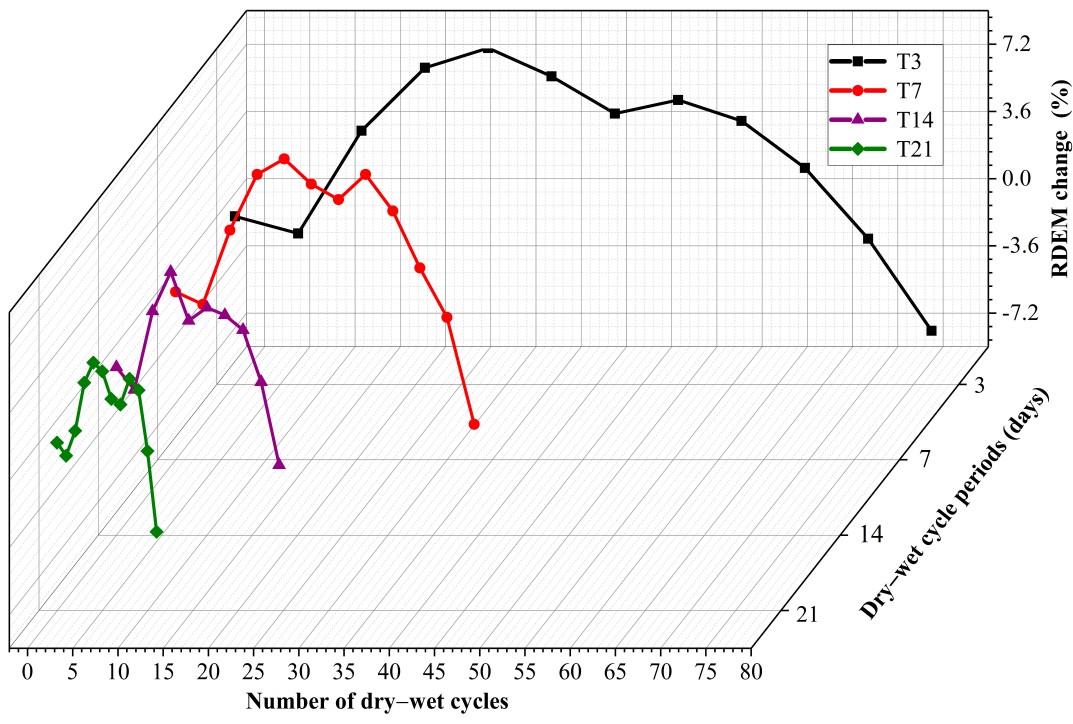

**Figure 8.** Variation of relative dynamic elastic modulus among different dry-wet cycle periods.

T3, T7, T14, and T21 peaked in relative dynamic elastic modulus after 28, 12, 6, and 4 dry–wet cycles, respectively. There are two aspects that caused the RDEM of the concrete to rise at this stage. On the one hand, hydration still happened inside the concrete and the soaking process was conducive to an increase in the dynamic elastic modulus; on the other hand, $SO_4^{2-}$ from the solution entered the interior of the concrete during the dry–wet cycles and reacted with the cement hydration products, producing gypsum and ettringite (among other products), which filled the internal pores and increased the density of the concrete. This inhibited the further entry of $SO_4^{2-}$, thus reducing the sulfate attack and improving the RDEM [32,33].

After peaking, the RDEM of each group began to decline—most rapidly for T7, with an average decrease of 0.7% per cycle. This is because as the attack continued, the combined effect of the concrete damage caused per cycle and the number of cycles sharply increased.

The shorter the dry–wet cycle period was, the more rapidly capillary adsorption would replenish $SO_4^{2-}$ in the wetting process, which reduced the concentration difference and humidity gradient inside the concrete during the drying process. This slowed down the $SO_4^{2-}$ migration to deep concrete and weakened the influence of a single cycle inside the concrete. Too frequent dry–wet cycles were not conducive to the reaction of $SO_4^{2-}$ with the cement hydration products, as discussed in detail below regarding the damage mechanism. With the further extension of the dry–wet cycle period, although the time available for $SO_4^{2-}$ to react with cement hydration products was increased, the interval between the wetting process was longer, and the supply of $SO_4^{2-}$ was limited. Therefore, the deterioration rate for the concrete gradually decreased.

**Table 5.** The dynamic elastic modulus in the experiment (Ed/×1000 MPa).

| Cycles | T3 | | | Average Value | Relative Value |
|---|---|---|---|---|---|
| | **1** | **2** | **3** | | |
| 0 | 43.8 | 43.2 | 43.8 | 43.6 | 0.00 |
| 7 | 43.4 | 42.9 | 43.3 | 43.2 | −0.92 |
| 14 | 45.1 | 45.5 | 46.2 | 45.6 | 4.59 |
| 21 | 47.6 | 45.9 | 47.7 | 47.1 | 7.95 |
| 28 | 47.6 | 47.6 | 47.4 | 47.5 | 9.02 |
| 35 | 47.4 | 46.4 | 46.8 | 46.9 | 7.49 |
| 42 | 46.5 | 45.6 | 45.9 | 46.0 | 5.50 |
| 49 | 46.4 | 46.1 | 46.4 | 46.3 | 6.19 |
| 56 | 46.4 | 45.0 | 46.1 | 45.8 | 5.12 |
| 63 | 44.8 | 44.5 | 44.9 | 44.7 | 2.60 |
| 70 | 43.4 | 43.0 | 42.8 | 43.1 | −1.22 |
| 77 | 40.8 | 41.2 | 40.8 | 40.9 | −6.12 |
| | T7 | | | | |
| | 1 | 2 | 3 | | |
| 0 | 44.7 | 44.3 | 44.5 | 44.50 | 0.00 |
| 3 | 43.7 | 44.3 | 44.6 | 44.20 | −0.67 |
| 6 | 45.6 | 46.0 | 46.3 | 45.97 | 3.30 |
| 9 | 47.6 | 47.6 | 46.7 | 47.30 | 6.29 |
| 12 | 47.4 | 46.8 | 48.8 | 47.67 | 7.12 |
| 15 | 46.4 | 47.0 | 47.8 | 47.07 | 5.77 |
| 18 | 45.9 | 46.4 | 47.8 | 46.70 | 4.94 |
| 21 | 47.3 | 47.5 | 47.1 | 47.30 | 6.29 |
| 24 | 46.3 | 46.5 | 46.5 | 46.43 | 4.34 |
| 27 | 44.1 | 45.0 | 46.1 | 45.07 | 1.27 |
| 30 | 44.0 | 43.8 | 43.9 | 43.90 | −1.35 |
| 33 | 41.2 | 41.5 | 41.3 | 41.33 | −7.12 |
| | T14 | | | | |
| | 1 | 2 | 3 | | |
| 0 | 44.0 | 44.0 | 44.2 | 44.07 | 0.00 |
| 2 | 42.5 | 43.6 | 44.5 | 43.53 | −1.22 |
| 4 | 45.4 | 45.5 | 45.3 | 45.40 | 3.02 |
| 6 | 46.3 | 46.5 | 46.2 | 46.33 | 5.14 |
| 8 | 44.9 | 45.2 | 45.4 | 45.17 | 2.49 |
| 10 | 45.2 | 45.5 | 45.7 | 45.47 | 3.17 |
| 12 | 44.3 | 45.4 | 46.2 | 45.30 | 2.79 |
| 14 | 43.9 | 45.0 | 45.8 | 44.90 | 1.88 |
| 16 | 43.0 | 43.8 | 44.4 | 43.73 | −0.76 |
| 18 | 41.3 | 41.6 | 42.4 | 41.77 | −5.23 |
| | T21 | | | | |
| | 1 | 2 | 3 | | |
| 0 | 42.9 | 42.3 | 42.9 | 42.70 | 0.00 |
| 1 | 43.1 | 41.0 | 43.1 | 42.40 | −0.70 |
| 2 | 44.1 | 42.0 | 42.8 | 42.97 | 0.62 |
| 3 | 44.5 | 43.1 | 44.6 | 44.07 | 3.20 |
| 4 | 44.4 | 43.9 | 45.3 | 44.53 | 4.29 |
| 5 | 43.6 | 44.3 | 45.1 | 44.33 | 3.83 |
| 6 | 44.0 | 42.9 | 44.2 | 43.70 | 2.34 |
| 7 | 43.6 | 43.4 | 43.7 | 43.57 | 2.03 |
| 8 | 44.2 | 44.0 | 44.3 | 44.17 | 3.43 |
| 9 | 44.0 | 43.8 | 43.9 | 43.90 | 2.81 |
| 10 | 43.1 | 42.1 | 42.3 | 42.50 | −0.47 |
| 11 | 41.5 | 40.3 | 40.2 | 40.67 | −4.76 |

### 3.4. Mass Change

The curves of the mass-loss rate with the number of cycles for different dry–wet periods are shown in Figure 9, the mass change in the experiment are listed in Table 6. It can be observed that the mass changed slowly, within a small range. The peak mass of the T21 group increased by 0.51%, while that of T3 increased by 0.29%, compared with the initial masses. The overall trend was an initial increase in mass and then a decrease; in the early stages, the weight of a specimen increased at a rate that increased with longer dry–wet cycles.

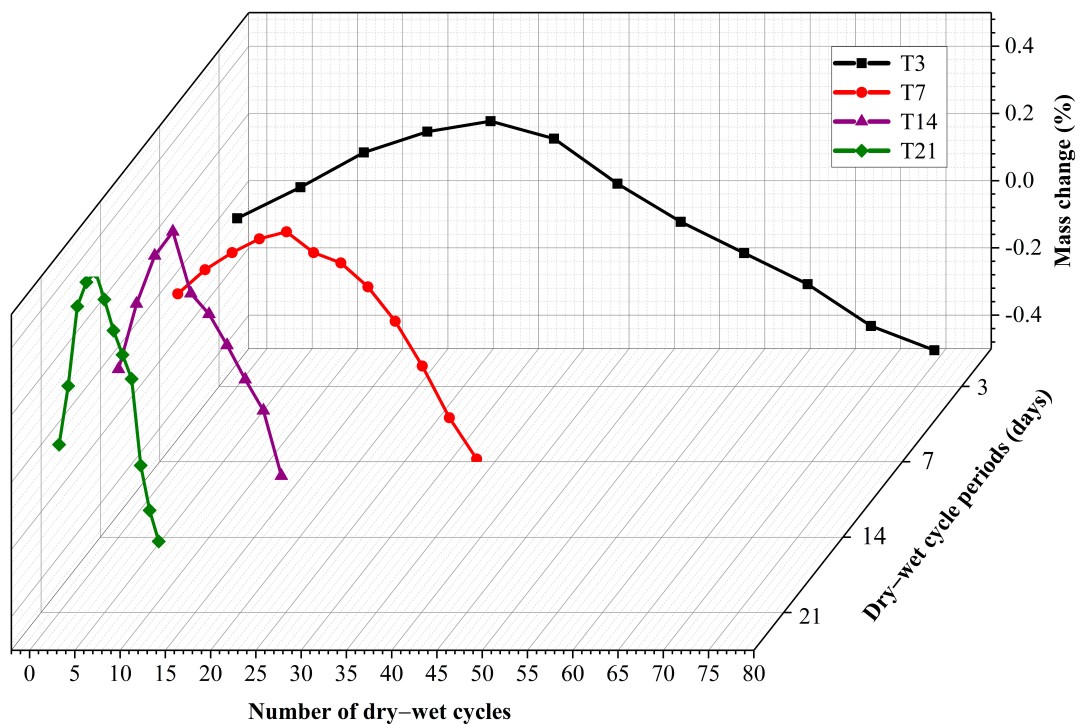

**Figure 9.** Variation of mass change among different dry-wet cycle periods.

However, the numbers of dry–wet cycles required for the rate of mass to peak were different; the shorter the dry–wet cycle period, the more cycles required. For example, T3's mass peaked at 28 cycles, while that of T14 peaked at only six. This difference again proves that the erosion intensity for a single cycle significantly increases with an increase in the cycle period.

The main reason for the mass increase of the concrete was that the $SO_4^{2-}$ in the solution entered the concrete in the wetting process and reacted with the cement hydration byproducts to form expansive products. During the drying process, the water evaporated and the $SO_4^{2-}$ concentration increased, speeding up the chemical reaction. Simultaneously, $Na_2SO_4$ crystallized out, increasing the quality of the concrete by filling its internal pores.

The rate of mass change rapidly decreased with an increase in dry–wet cycles. It can be seen from Figure 6 that the average single-cycle mass-loss rate for T21 was the largest, at 0.11%. However, the overall decrease for T7 was the largest, at about 0.49% compared with the initial value.

**Table 6.** The mass change in the experiment (kg).

| Cycles | T3 | | | Average Value | Relative Value |
|---|---|---|---|---|---|
| | **1** | **2** | **3** | | |
| 0 | 9.690 | 9.680 | 9.690 | 9.687 | 0.00 |
| 7 | 9.703 | 9.686 | 9.698 | 9.696 | 0.09 |
| 14 | 9.717 | 9.690 | 9.710 | 9.706 | 0.20 |
| 21 | 9.723 | 9.695 | 9.719 | 9.712 | 0.26 |
| 28 | 9.725 | 9.700 | 9.721 | 9.715 | 0.29 |
| 35 | 9.721 | 9.694 | 9.715 | 9.710 | 0.24 |
| 42 | 9.707 | 9.680 | 9.703 | 9.697 | 0.10 |
| 49 | 9.696 | 9.672 | 9.691 | 9.686 | −0.01 |
| 56 | 9.688 | 9.664 | 9.679 | 9.677 | −0.10 |
| 63 | 9.680 | 9.656 | 9.668 | 9.668 | −0.20 |
| 70 | 9.668 | 9.645 | 9.655 | 9.656 | −0.32 |
| 77 | 9.660 | 9.639 | 9.648 | 9.649 | −0.39 |
| | T7 | | | | |
| | 1 | 2 | 3 | | |
| 0 | 9.753 | 9.751 | 9.795 | 9.766 | 0.00 |
| 3 | 9.770 | 9.750 | 9.800 | 9.773 | 0.07 |
| 6 | 9.763 | 9.762 | 9.808 | 9.778 | 0.12 |
| 9 | 9.768 | 9.769 | 9.810 | 9.782 | 0.16 |
| 12 | 9.771 | 9.768 | 9.813 | 9.784 | 0.18 |
| 15 | 9.766 | 9.762 | 9.807 | 9.778 | 0.12 |
| 18 | 9.761 | 9.759 | 9.805 | 9.775 | 0.09 |
| 21 | 9.755 | 9.752 | 9.798 | 9.768 | 0.02 |
| 24 | 9.748 | 9.743 | 9.784 | 9.758 | −0.08 |
| 27 | 9.733 | 9.729 | 9.773 | 9.745 | −0.22 |
| 30 | 9.718 | 9.717 | 9.755 | 9.730 | −0.37 |
| 33 | 9.712 | 9.703 | 9.740 | 9.718 | −0.49 |
| | T14 | | | | |
| | 1 | 2 | 3 | | |
| 0 | 9.803 | 9.806 | 9.700 | 9.770 | 0.00 |
| 2 | 9.821 | 9.828 | 9.717 | 9.789 | 0.19 |
| 4 | 9.836 | 9.836 | 9.736 | 9.803 | 0.34 |
| 6 | 9.845 | 9.846 | 9.738 | 9.810 | 0.41 |
| 8 | 9.826 | 9.827 | 9.724 | 9.792 | 0.23 |
| 10 | 9.820 | 9.822 | 9.716 | 9.786 | 0.16 |
| 12 | 9.811 | 9.813 | 9.706 | 9.777 | 0.07 |
| 14 | 9.802 | 9.803 | 9.695 | 9.767 | −0.03 |
| 16 | 9.789 | 9.790 | 9.696 | 9.758 | −0.12 |
| 18 | 9.769 | 9.775 | 9.673 | 9.739 | −0.32 |
| | T21 | | | | |
| | 1 | 2 | 3 | | |
| 0 | 9.720 | 9.720 | 9.720 | 9.720 | 0.00 |
| 1 | 9.710 | 9.710 | 9.790 | 9.737 | 0.17 |
| 2 | 9.733 | 9.735 | 9.812 | 9.760 | 0.41 |
| 3 | 9.739 | 9.743 | 9.820 | 9.767 | 0.48 |
| 4 | 9.743 | 9.744 | 9.822 | 9.770 | 0.51 |
| 5 | 9.734 | 9.738 | 9.813 | 9.762 | 0.43 |
| 6 | 9.727 | 9.726 | 9.805 | 9.753 | 0.34 |
| 7 | 9.719 | 9.722 | 9.801 | 9.747 | 0.27 |
| 8 | 9.712 | 9.714 | 9.790 | 9.739 | 0.20 |
| 9 | 9.689 | 9.691 | 9.763 | 9.714 | −0.06 |
| 10 | 9.676 | 9.675 | 9.753 | 9.701 | −0.20 |
| 11 | 9.662 | 9.666 | 9.747 | 9.692 | −0.29 |

### 3.5. Microstructural Analysis by SEM

Figure 10 shows the microstructural morphology of the specimens before and after the sulfate dry–wet cycles. Figure 10a shows that the concrete sample without sulfate attack had better compactness and plenty of flocculated fibrous calcium silicate hydrate (C-S-H), and some acicular ettringite (Aft) can be observed. In addition, some micropores and microcracks are also observable. Fly ash contains many active ingredients, such as $SiO_2$ and $Al_2O_3$. On the one hand, a secondary hydration reaction occurred between the fly ash and cement hydration product calcium hydroxide (CH), which generated the gel substance calcium silicate hydrate (C-S-H) and improved the composition of the cementitious materials in the concrete and the compactness of the cement matrix [34]. On the other hand, the presence of active $Al_2O_3$ might increase the content of sulfoaluminate hydrate in the hydration product.

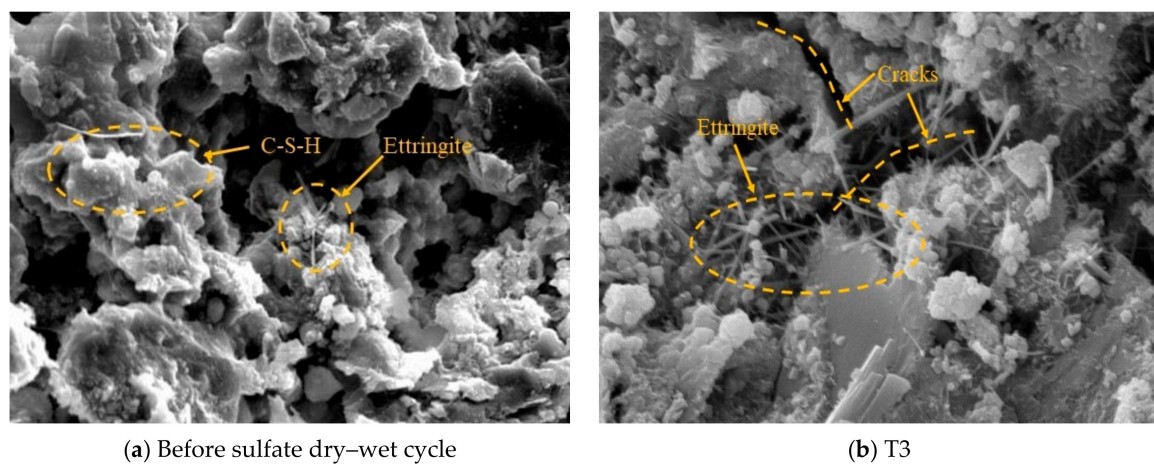

(**a**) Before sulfate dry–wet cycle                (**b**) T3

**Figure 10.** The microstructure before and after the sulfate dry–wet cycle for 84 days. (**a**) Before sulfate dry–wet cycle; (**b**) T3.

After 28 cycles, the attack products were greatly increased in amount compared to the non-attacked specimens. A large number of needle-shaped ettringite crystals could be found in T3, and hydrated calcium silicate gel (C-S-H) was observed in a specific area. The specimen compactness was enhanced because erosion products filled the concrete's micropores and microcracks. There was a rapid increase in flexural strength, specimen mass, and dynamic elastic modulus. Increasing the dry–wet cycle period significantly changed the number and volume of these products, such as ettringite, enhancing the effects in the initial stage of concrete attack and the terminal deterioration.

As shown in Figure 11, many needle-like prismatic ettringite (AFt) crystals and much plate-like gypsum could be observed inside the shallow area (15 mm), but more so in T7 than T3. This indicates that the erosion intensity continued to increase with the dry–wet cycle period. With an increase in the attack depth (30 mm), the amounts of attack products continuously decreased. The ettringite was mainly in the shape of needles and had a smaller volume, as shown in Figure 12a. Therefore, the damage to the concrete caused by the dry–wet sulfate cycles gradually decreased. Upon augmenting the dry–wet cycle period, in the deep area (30 mm), large gypsum crystals formed in T21 and ettringite was distributed in the form of small and fine needles in the flocculent C-S-H that had not been completely eroded (Figure 12b). The cementation became fibrous, the ettringite (AFt) distribution was uneven, and there were cracks, but the amounts of attack products slightly increased compared to T3. This means that increasing the interval between wetting processes restricted the supply of $SO_4^{2-}$, thereby reducing the generation of erosion byproducts and, consequently, the erosion intensity. Meanwhile, the ability of the dry–wet process to affect the deeper layers of the concrete also decreased. Therefore, the depth of influence inside the specimen was ultimately limited, restricted by the period of the dry–wet cycle.

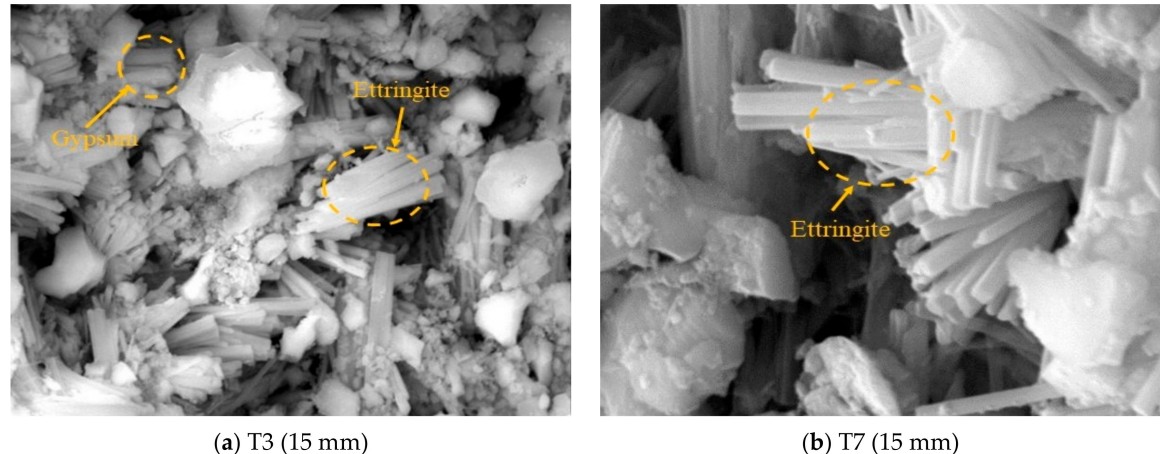

(**a**) T3 (15 mm)     (**b**) T7 (15 mm)

**Figure 11.** The microstructures of the specimens after sulfate dry–wet cycles for 210 days. (**a**) T3; (**b**) T7.

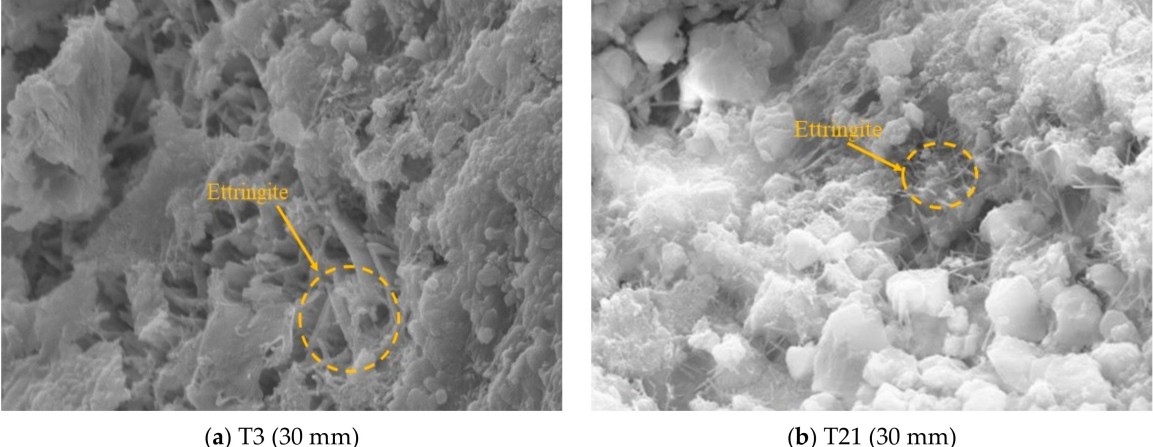

(**a**) T3 (30 mm)     (**b**) T21 (30 mm)

**Figure 12.** The microstructures of the specimens after sulfate dry–wet cycles for 210 days. (**a**) T3; (**b**) T21.

### 4. Conclusions

In this study, concrete exposed to a sulfate solution with different dry–wet cycle periods was tested. The flexural strength, dynamic elastic modulus, and mass of the specimens were measured, and SEM images of different corrosion processes were recorded. The following conclusions can be drawn:

(1) The changes in concrete performance with different dry–wet cycles were essentially consistent. Within a certain range, the erosion of concrete became more apparent upon prolonging the dry–wet cycles. The relative flexural strength and relative dynamic modulus of the T3, T7, T14, and T21 specimens peaked at 28, 12, 6, and 4 dry–wet cycles, respectively. After 252 days of sulfate dry–wet cycles, the flexural strength of the single-cycle specimens decreased by 1.05%, 2.7%, 4.2%, and 5.6% on average, respectively.

(2) Regarding the microstructure, increasing the dry–wet cycle period gradually increased the depth of influence of the sulfate attack inside the concrete, along with the number and volumes of erosion products at a given depth. These results well validate the variation law of macroscopic performance. However, for a given dry–wet cycle period, the content of erosion products gradually decreased with increasing depth.

(3) The excessive extension of the dry–wet cycle period has little effect on the deterioration of concrete's performance; thus, such extension is not advised. According to the test results, the strengths and RDEMs of T14 and T21 showed similar variations. The microstructural analysis illustrated that the number and volume of attack products

inside the specimen showed trends of decreasing with increasing dry–wet cycle periods. Therefore, based on the strength and depth of single-cycle erosion, it is advisable to set the cycle period to 7–14 days when testing sulfate dry–wet attack on concrete under natural dry conditions.

**Author Contributions:** Conceptualization, J.-J.G. and K.W.; methodology, J.-J.G., P.-Q.L. and K.W.; analysis, J.-J.G., P.-Q.L., C.-L.W. and K.W.; investigation, J.-J.G., P.-Q.L. and C.-L.W.; resources, K.W.; writing and editing, J.-J.G., K.W. and P.-Q.L.; visualization, P.-Q.L.; supervision, C.-L.W.; project administration, K.W.; funding acquisition, J.-J.G. and C.-L.W. All authors have read and agreed to the published version of the manuscript.

**Funding:** This research was funded by the National Natural Science Foundation of China, grant number 51879244 and 51679221, and the Program for Innovative Research Team (in Science and Technology) at the University of Henan Province, grant number 20IRTSTHN009.

**Institutional Review Board Statement:** Not applicable.

**Informed Consent Statement:** Not applicable.

**Data Availability Statement:** Not applicable.

**Acknowledgments:** The financial support from the Zhengzhou University Postgraduate Independent Innovation Project and Zhengzhou University is also acknowledged.

**Conflicts of Interest:** The authors declare no conflict of interest.

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
