# Peer review of "Effect of Dry–Wet Cycle Periods on Properties of Concrete under Sulfate Attack"

_applsci, doi:10.3390/app11020888_

Round 1
Reviewer 1 Report
The manuscript describes an experimental study of wet-drying cycles on properties of mortars subjected to sulphate attact. The investigated properties are flexural strength, dynamic modulus of elasticity and mass. Morover, some qualitative SEM analysis of hydration products developement and microscraking was performed. The paper has clear goals, is concise, understandable and interesting for the scientific oriented reader. The English language of the manuscript needs very cautious and deep review. It is a pitty that the experimental programme was designed only for one mortar mix. It would greately improve the content if for instance different fly ash additions would be considered. Apart from this I have some more specific remarks listed below:
- There are smany typos to be corrected like "5% Na2SO4 solution with a mass fraction of was prepared as the test erosion solution." or "Consequently, load on the specimen at 0.06 MPa/s until broking. Then record the failure pressure and the fracture position of the lower edge of the specimen." or "The measuring point was located in the middle of one side of the forming surface of the specimen that no visible hole or crack" "The overall trend was to strengthen firstly and then decreased."
- Why the w/b ratio was chosen at a peculiar level of w/b=0.58?
- "The main reason for the emergence of rebound is caused by the swelling products such as ettringite generated by sulfate attack filling the concrete pores and cracks resulted from early attack" This sentence should be supported either by the Authors own results or some literature references.
- How many samples were tested for mass, flexural strangth and modulus? It seems that only single samples were tested from the presented figures. If yes, could the Authors mention the possible statistical distribution on the obtained results? If not, please present the individual results, mean values and coefficient of variation in tables.
- It would be interesting to present Figures 7-9 as 3D graphs including the wet-drying cylces period on the third axis. It would help to understand the observed trends.
Taking into considerations all the overmentioned remarks I recommend reconcidering paper's acceptance after a major review with a special focus on improving English.
Reviewer 2 Report
An interesting article, well written, well edited.
It deals with the important practical issue of sulfate corrosion of concrete.
It requires a slight correction.
The ordinates in all the graphs in Figure 7 should vary in the same range, for example from -10% to +10%. This will allow for a better illustration of the differences for the tested dry wet cycle periods.
This note also applies to the diagrams in Figures 8 and 9.
Round 2
Reviewer 1 Report
The Authors have addressed all my comments and remarks. I recommend accepting the manuscript in the present form.